# Patterns of Centennial-to-Millennial Holocene Climate Variation in the North American Mid-Latitudes

Bryan N. Shuman[1]

5    [1]Department of Geology and Geophysics, University of Wyoming, Laramie, WY, 82071, USA

*Correspondence to*: Bryan N. Shuman (bshuman@uwyo.edu)

**Abstract.** Noise in Holocene paleoclimate reconstructions can hamper detection of centennial-to-millennial climate variations and diagnoses of the dynamics involved. This paper uses multiple ensembles of reconstructions to separate signal and noise and determine what, if any, centennial-to-millennial variations influenced North America during the past 7000 yr. To do so, ensembles of temperature and moisture reconstructions were compared across four different spatial scales: multi-continent, regional, sub-regional, and local. At each scale, two independent multi-record ensembles were compared to detect any centennial-to-millennial departures from the long Holocene trends, which correlate more than expected from random patterns. In all cases, the potential centennial-to-millennial variations had small magnitudes. However, at least two patterns of centennial-to-millennial variability appear evident. First, large-scale variations included a prominent Mid-Holocene anomaly from 5600-5000 YBP that increased mean effective moisture and produced temperature anomalies of different signs in different regions. The changes shifted the north-south temperature gradient in mid-latitude North America with a pattern similar to that of the North Atlantic Oscillation (NAO). Second, correlated multi-century (~350 yr) variations produce a distinct spectral signature in temperature and hydroclimate records along the western Atlantic margin. Both patterns differ from random variations but they express distinct spatiotemporal characteristics consistent with separate controlling dynamics.

**Graphical Abstract**

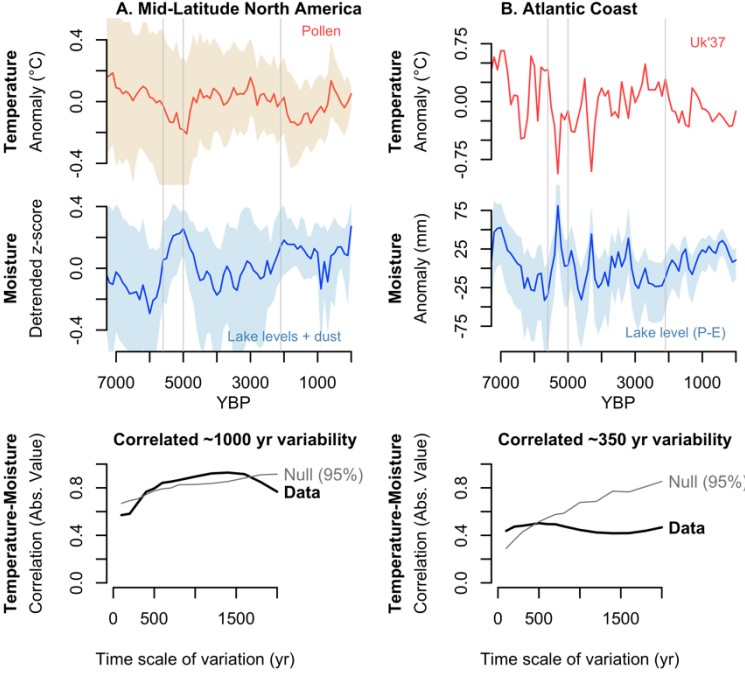

## 1 Introduction

A prominent gap in the conceptual and empirical understanding of the full spectrum of climate variation exists at centennial-to-millennial (cen-mil) scales, particularly for warm climate states (Crucifix et al., 2017; Hernández et al., 2020; Wanner et al., 2008). Cen-mil variations exceed the scales of direct observation and yet are short relative to the resolution of many geological records. Climate models can also fail to sufficiently resolve this scale of climate variation, which is particularly important for anticipating regional changes (Hébert et al., 2022; Laepple et al., 2023). The resulting gap in understanding has been recognized for decades (Saltzman, 1982) and continues to thwart efforts to understand 'low-frequency' components of change during the Common Era and other times (Ault et al., 2013). The Holocene Epoch offers a key to this 'missing' scale because cen-mil variability across a broad continuum (Hernández et al., 2020; Huybers and Curry, 2006; Mayewski et al., 2004; Wanner et al., 2008; Hébert et al., 2022) likely played an important role in shaping ecological, geomorphic, and human history during the past 11,700 years (Fletcher et al., 2013; Shuman et al., 2019; deMenocal, 2001).

Cen-mil variation during the Holocene must have arisen from the interaction among: 1) deterministic changes, such as energy balance responses to seasonal insolation trends; 2) chaotic dynamics, such as the fluid behaviors of the atmosphere and oceans and complex biosphere feedbacks; and 3) stochastic events and variability, such as is common on interannual to decadal scales. Non-linear, probabilistic, or transient cen-mil dynamics may have followed the Holocene's orbital and greenhouse gas changes (Claussen et al., 1999; Saltzman, 1982; Wan et al., 2019), autoregressive solar variations (Renssen et al., 2006), and stochastic volcanic eruptions (Kobashi et al., 2017). However, intrinsic, unforced cen-mil variability arising from atmospheric, ocean and sea-ice dynamics may have been equally as significant (Ault et al., 2018) with ocean variability driving regional change even over the continents (Hébert et al., 2022). Simulations produce repeated unforced millennial variations, which are similar in magnitude to the effects of early Holocene meltwater forcing and are correlated over the northern continents and oceans (Marsicek et al., 2018; Wan et al., 2019). Some variations likely centered on the North Atlantic (Thornalley et al., 2009; Anchukaitis et al., 2019) and the tropical Pacific (Karnauskas et al., 2012) with the potential for far-reaching spatial expressions like those expressed at other time scales. Biosphere feedbacks may have also triggered state shifts against the backdrop of other Holocene variability (Claussen et al., 1999; deMenocal et al., 2000). Importantly, internal variability may generate variations of opposing sign in different regions with areas of change and no change in close proximity (Shuman et al., 2023).

A daunting breadth of variation among individual records matches the wide range of possible drivers (Mayewski et al., 2004; Wanner et al., 2008), but signal-to-noise ratios are small. Analytical or calibration uncertainties of ~2°C (e.g., Russell et al., 2018; Williams and Shuman, 2008; Martínez-Sosa et al., 2021) often dwarf the expected magnitudes of cen-mil temperature variation during the Holocene (~0.5 °C in CCSM3 TRACE simulations; Marsicek et al., 2018; Wan et al., 2019). At the same time, reconstruction techniques, age uncertainties, and averaging across multiple records can reduce the apparent amplitude of cen-mil climate variations (Hébert et al., 2022). Furthermore, slow Earth system components like the ocean can integrate

stochastic interannual variability to produce autoregressive variations at cen-mil scales (Huybers and Curry, 2006), but oscillations in many paleoclimate datasets may be noise generated by smoothing or other integrating processes. Statistical

transformations (e.g., curve fitting), sampling effects (e.g., homogenized samples spanning decades), and a wide range of environmental filtering processes such as sediment mixing, lake residence times, or slow forest turn-over add temporal autocorrelation generating oscillations from white noise via the Slutzky-Yule Effect (Slutzky, 1937). Diagnosing the continuum of weak signals and their complex interactions amid the full array of noise and uncertainties creates a unique challenge for studying cen-mil variability but is essential for determining the origins of changes that may modify future trends,

particularly at regional and finer spatial scales.

The goal of developing a "coherent, falsifiable narrative" (cf. Bender, 2013) may help: what signals are coherent among independent datasets? Are they falsifiably different from null expectations about noise? This paper applies these questions to cen-mil variations during the past 7000 years, after ice sheet influences diminished. The analyses focus on variations previously

observed at four spatial scales centered on eastern mid-latitude North America to describe the types of Holocene variations expressed in the northern mid-latitudes. Cen-mil variations in this region include long-term dynamics similar to the interannual North Atlantic Oscillation (NAO; Olsen et al., 2012; Orme et al., 2021; Shuman et al., 2023), correlations among North American droughts and Atlantic temperatures (Shuman et al., 2019; Shuman and Burrell, 2017; Anchukaitis et al., 2019), and possible large-scale temperature variations, consistent in frequency and magnitude with unforced variability simulated by

models (Marsicek et al., 2018; Herzschuh et al., 2023). To compare the expression of cen-mil variability across spatial scales, dissimilar geographic regions, and different climate variables, the analyses used here focus on a spatial hierarchy of paired multi-record ensembles. The scales span from hemispheric variations that extend well beyond North America to sub-regional and local variations that differ among areas of the continent:

- **Multi-continent cen-mil signals** in both a temperature ensemble derived from European and North American pollen
records and an ensemble of water temperature records (Marsicek et al., 2018)(Fig. 1A);

- **Regional cen-mil signals** in networks of temperature and moisture records spanning mid-latitude North America (Shuman and Marsicek, 2016)(Fig. 1B);

- **Sub-regional cen-mil signals** in reconstructions from the central and northeastern subregions of mid-latitude North America (Shuman and Burrell, 2017; Shuman and Marsicek, 2016)(Fig. 1B); and

- **Site-level cen-mil signals** within the sub-regions, including individual temperature and hydroclimate records (Shuman and Burrell, 2017; Shuman and Marsicek, 2016)(individually labelled sites, Fig. 1B).

The analyses build upon the detection of cen-mil signals that were cross-validated across sites and multiple lines of evidence in the northeast U.S. (Shuman et al., 2019), and compares them to evidence of different cen-mil signals at the larger regional

(Shuman and Marsicek, 2016) and continental scales (Marsicek et al., 2018). To help evaluate potential cen-mil variations, correlations among paleoclimate reconstructions were evaluated at different frequencies and compared against the level of

spurious correlations in random data series (Reschke et al., 2019). Random series reveal signal characteristics, such as the strength of signal magnitudes and correlations among multi-records ensemble means, which could help distinguish between cen-mil noise and climatically significant variations (Fig. 2). To further diagnose the variations, spatial patterns associated

with temperature change at 5500-5000 YBP are also evaluated; the interval represents the largest rate of change in pollen-inferred temperatures, lake-level changes, and stable isotope records from mid-latitude North America (Shuman and Marsicek, 2016) and includes a potential step shift in hemispheric temperatures (Marsicek et al., 2018). The mid-Holocene changes may relate to commonly detected anomalies in many North American records and could have characteristics (i.e., changes in the latitudinal temperature gradient over North America) similar to the NAO at interannual scales (Hurrell et al., 2003; Folland et

al., 2009; Shuman et al., 2023). Together, the results demonstrate a set of cen-mil changes over the past 7000 years, which correlate among ensembles of temperature and moisture reconstructions while differing among regions and scales likely because they represent multiple dynamics.

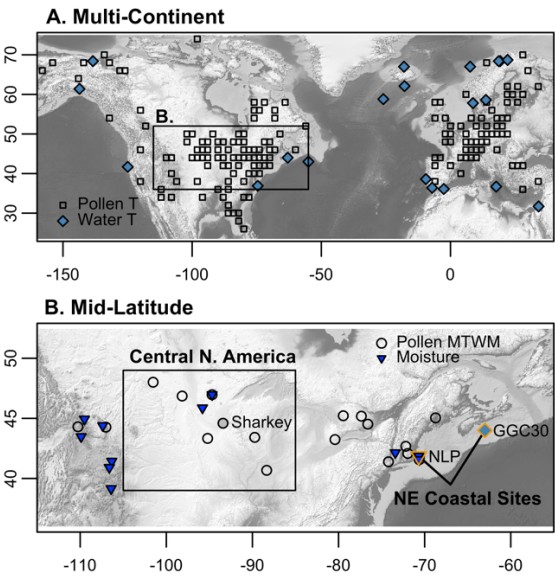

**Figure 1 Locations of paleoclimate reconstructions incorporated into ensembles of A) continental-scale mean annual temperatures ('pollen T', squares; Marsicek et al., 2018) and water temperatures ('water T', blue circles; from Marcott et al., 2013) and B) mid-latitude, central, and northeast (NE) coastal mean temperatures of the warmest month (MTWM, circles; Shuman and Marsicek 2016) and moisture availability (blue triangles; Shuman and Marsicek 2016). The mid-latitude ensembles include all pollen-inferred MTWM and moisture records shown in B; the central North America ensembles include only sites in the inset box; and the NE**
**coastal comparison uses the two sites outlined in orange (New Long Pond, NLP, Newby et al., 2014; OCE326-GGC30, a water temperature site in A from Sachs, 2007). In the central region, the MTWM mean excludes and is compared to Sharkey Lake, MN (gray).**

## 2 Methods

For this analysis, cen-mil variations in Holocene paleoclimate timeseries are defined as variations evident in paleoclimate reconstructions that have been binned at 100-yr time steps and then detrended using a Gaussian filter with a 6000-yr window (see Reschke et al., 2019). They are considered significant, and therefore climatically meaningful, if they are correlated in independent paleoclimate reconstructions more than expected from surrogate timeseries, randomly generated as power-law time series with the same spectral slope over the 100-2000 year band of variation as determined from the paleoclimate reconstructions (following Reschke et al., 2019).

All of the paleoclimate datasets examined here have been previously published. Additional data may help further evaluate the patterns involved, but this paper aims to interrogate and compare the patterns over the past 7000 years in these specific representative datasets to generate an initial evaluation of possible patterns.

At the largest ("multi-continent") scale (Fig. 1A), Marsicek et al. (2018) used 642 fossil pollen records, which were averaged over a grid, to reconstruct Holocene mean annual temperature (MAT) trends across Europe and North America (grid points used for the reconstruction are shown as squares in Fig. 1A). They showed cen-mil variations, which correlated across the two continents, between random sub-sets of the reconstructions, and with those detected in an ensemble of geochemical and chironomid-inferred water temperature records from marine and lake cores in the North America-Europe region (blue circles, Fig. 1A). The mean European and North American MAT reconstruction and the water temperature ensemble are used here as two independent records of possible large-scale cen-mil variations, which may have had consequences in mid-latitude North America.

At the mid-latitude (regional) scale (Fig. 1B), Shuman and Marsicek (2016) synthesized 40 well-resolved paleoclimate records from across mid-latitude North America. They produced independent reconstructions of 1) the regional mean temperature of the warmest month (MTWM), averaged from 16 pollen-inferred records (circles, Fig. 1B), and 2) effective moisture, reconstructed using 9 lake-level and 2 dust records averaged as z-scores (blue triangles, Fig. 1B). The cen-mil variations in the independent MTWM and moisture reconstructions have not been previously compared. To evaluate the spatial patterns involved in the most prominent variation, an unusually cool period from 5600-5000 YBP, the MTWM dataset is also split here into two sub-sets for comparison based on the direction of the previously identified mid-Holocene temperature change (Shuman and Marsicek, 2016; Shuman et al., 2023). Sites were selected by dividing the records into two groups based on the mean difference in MTWM between two 600-yr periods (5600-5000 minus 4700-4100 YBP) (Fig. 3C; Shuman and Marsicek, 2016). The mean change across the 600-yr periods was used to minimize any artificial alignment of noise in the data caused by randomly splitting the data based on changes at a narrow, fixed point in time. Ensemble means with uncertainty distributions for each group were produced by randomly selecting and averaging five individual pollen-inferred MTWM records, which

were selected from each group with replacement 100 times. A comparison is also made with geochemical data from the two areas.

At finer scales, Shuman and Marsicek (2016) also produced sub-regional temperature and moisture reconstructions based on the averages of records from geographically distinct areas, including in central North America and the northeast coast (Fig. 1B). The resulting MTWM and moisture reconstructions from central North America are used here to examine cen-mil variation in a mid-continent sub-region (inset box, Fig. 1B). The central sub-region ensemble incorporates 6 pollen-inferred MTWM reconstructions from North Dakota, Minnesota, Iowa, Wisconsin, and Illinois; moisture history was reconstructed from 2 dust records from Minnesota. One centrally located MTWM reconstruction based on fossil pollen data from Sharkey Lake, Minnesota (Camill et al., 2003) is excluded and compared with the sub-regional ensemble to confirm the sub-regional pattern and represent the local scale expression of the cen-mil variations (gray circle, Fig. 1B).

For comparison, and to further evaluate cen-mil variation at the finest, local scale, two individual records from the northeast (NE) coast of North America are used (orange outlined symbols, Fig. 1B). The records are representative of the NE coastal sub-region and have been identified as having correlated cen-mil temperature and moisture variations (Shuman et al., 2019; Shuman and Burrell, 2017; Shuman and Marsicek, 2016). The alkenone-inferred sea-surface temperature (SST) reconstruction from the Scotian Margin, core OCE326-GGC30 (Sachs, 2007) is used to capture sub-regional temperature variations; its temperature variations correlate with those represented by Shuman and Marsicek (2016)'s regional MTWM reconstruction based on five fossil pollen sites from the northeast U.S. The lake-level reconstruction from New Long Pond, Massachusetts, is highly correlated with similar reconstructions from other near-by lakes (Newby et al., 2014; Shuman and Burrell, 2017) and with pollen-inferred precipitation changes at five fossil pollen sites from same area (Shuman et al., 2019). Synchrony analyses of the calibrated radiocarbon dates used to constrain the paleoshoreline deposits support the cen-mil details of the lake-level reconstruction by confirming synchronous changes in multiple lakes (Newby et al., 2014); and the pollen-inferred precipitation reconstructions accurately inform simulations of the stratigraphic record of lake-level changes using forward modelling approaches (Shuman et al., 2019). For this reason, the analysis here focuses on temperature-moisture correlations between individual sites (including local noise but avoiding any effects of multi-site averaging), but is representative of sub-regional means from the coastal NE.

**1.1 Analyses**

To extract the cen-mil component of the eight datasets, curves were fit to the long-term trends using a Gaussian filter with a 6000-yr window using the *smoother* package in R (R Core Development Team, 2020); analyses of the cen-mil variations then use the detrended time series derived from the residuals of the Gaussian smoother. The previously published confidence intervals from each of the eight datasets (Marsicek et al., 2018; Shuman and Marsicek, 2016; Shuman and Burrell, 2017) were also detrended by subtracting the same Gaussian filtered trends. Because most of the reconstructions represent multi-record

ensembles, they have been previously interpolated and binned at 100- or 50-yr time steps to combine the unevenly spaced individual data. Here, the 100-yr time steps are used, given a mean sample resolution of 120 yr/sample for the mid-latitude pollen records (Shuman and Marsicek, 2016). The resulting timeseries are smoothed and then detrended to focus on multi-century and longer patterns of variation.

Once detrended, Pearson's product moment correlation coefficients were calculated for comparison with null distributions. To do so, the CorQuantilesNullHyp function in the *corit* package in R was used to calculate the Pearson's product moment correlation coefficients at specific timescales of variation with periods ranging from 100-2000 years for comparison with null distributions of correlations among 1000 random surrogate power-law timeseries with the same spectral slopes as the detrended reconstructions (Reschke et al., 2019). The independent data series for each scale or region were also compared using generalized least squares regression (gls in the *mgvc* package in R) to account for correlated errors by assuming a first order moving average structure in the residuals; gls slopes significantly different than zero provide additional evidence of a significant correlation. Spectral analyses were performed on the detrended cen-mil residuals using the *multitaper* package in R to compute multitaper spectral estimates (Percival et al., 1993). All of the analyses were applied only to 7000 years of data, although longer segments of the time series are also plotted.

**1.2 Demonstration of random effects**

The low signal-to-noise ratios and the timescale of cen-mil variation create particular challenges for detecting robust signals. Null expectations can help assess whether analyses show significant patterns (Reschke et al., 2019). To visually demonstrate the problem, 1000 random number series representing 11,000 simulated years were generated. For each series, 11,000 numbers were drawn from a normal distribution to produce time series of white noise. Loess curves with a 500-yr span were then fit to each series to generate Slutzky-Yule oscillations by smoothing the random time series to produce cen-mil variations similar to those that might appear in paleoclimatological datasets. Finally, each series was re-sampled randomly 110 times, consistent with the temporal frequency of sub-samples analyzed from many Holocene sediment cores. A second, comparable dataset with a "signal" was generated by selecting one of the random series to serve as a pseudo-signal and averaging it with each of the other 999 individual random series. The result produces a second set of time series that contain 50% signal and 50% random variation, which aim to mimic the noisy appearance of any real cen-mil variations in a set of paleoclimate records; i.e., these time series contain a 'real' shared signal, but it is partially obscured by cen-mil noise.

# 3 Results

## 3.1 Characteristics of Simulated Random Variation

Simulating random time series and then smoothing over 500-yr windows and re-sampling at intervals consistent with paleoclimatologic records produces time series with cen-mil variation similar to that observed in many Holocene datasets (Fig. 2A, particularly if long trends were added to these series). The random data series include eye catching, but spurious, alignments of variations (e.g., negative correlations in the bottom two series, Fig. 2A) and illustrate two important features that may serve as null expectations for evaluating actual paleoclimate time series. First, averages of multiple records (such as the six shown in Fig. 2A) produce time series with small amplitudes of variation compared to the individual time series because the random fluctuations interfere with each other. The ratio of the standard deviation of the mean time series (bold, Fig. 2A) and that of 1000 individual random series ranges from 0.27-0.41 (median: 0.33) because the individual series have larger standard deviations than the dampened means when random variations cancel each other in the mean. Second, Pearson's correlation coefficients (r) among the random time series center around zero; in this example, the 95% distribution of 1000 simulated time series equaled -0.29 to 0.30.

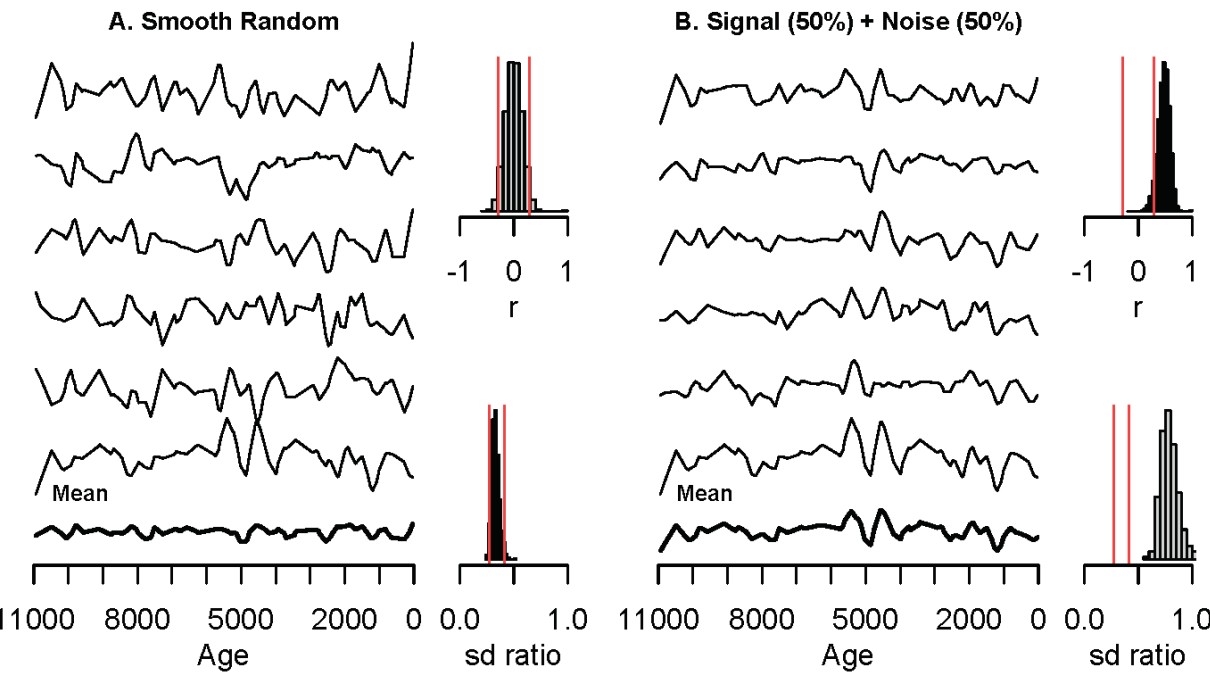

**Figure 2 Simulated time series representing A) six white-noise series that have been smoothed to produce spurious oscillations with ~500-yr periods and B) six series as in A, but where the random series has been averaged with a common signal taken from the lower-most series in A. Bold lines at the bottom represent the means of the six series in each case. Histograms show (at the top) the Pearson's product moment correlation coefficients, r, between the means and 1000 similarly generated random (A) or non-random**

**(B) time series, and (at the bottom) the ratio of the standard deviation (sd) of each mean and each of the 1000 simulated series. Red lines show the 95% distributions of the values for the random series.**

When a signal is incorporated into the simulated series by averaging the random variations and a common set of fluctuations (Fig. 2B), the results differ from the two null expectations: 1) multi-record means retain a similar apparent amplitude of variation to the individual datasets (standard deviation ratio >0.5) and 2) most of the correlations among individual time series

fall outside the random distribution. The six-record mean in Fig. 2B preserves a ratio of standard deviations equal to 0.66-0.97 (median: 0.77) and correlates well (r = 0.96) with the introduced signal (which is the lower-most record in both panels of Fig. 2).

The simulated mean illustrates the potential for averaging ensembles of records to help retrieve weak signals, difficult to

235 discern in individual records. Reconstructions, particularly those averaged across sites, reduce the amplitude of any cen-mil signals (as demonstrated by standard deviation ratios <1 in Fig. 2B), but averages of multiple records help to reduce random noise and enhance the correlation between the true signal and the ensemble reconstruction (r > 0.5 in Fig. 2B, but <0.4 in Fig. 2A). By contrast, noise introduced into each of the individual time series confounded the signals. For example, two event peaks at ca. 5500 and 4500 years in the 'signal' do not appear in all of the series in Fig. 2B, and the absence of one versus the other

can create a misleading assessment of a single, asynchronous event depending on the records used. The ensemble mean, however, successfully represents the underlying signal with both peaks (compare the bottom two curves, Fig. 2B).

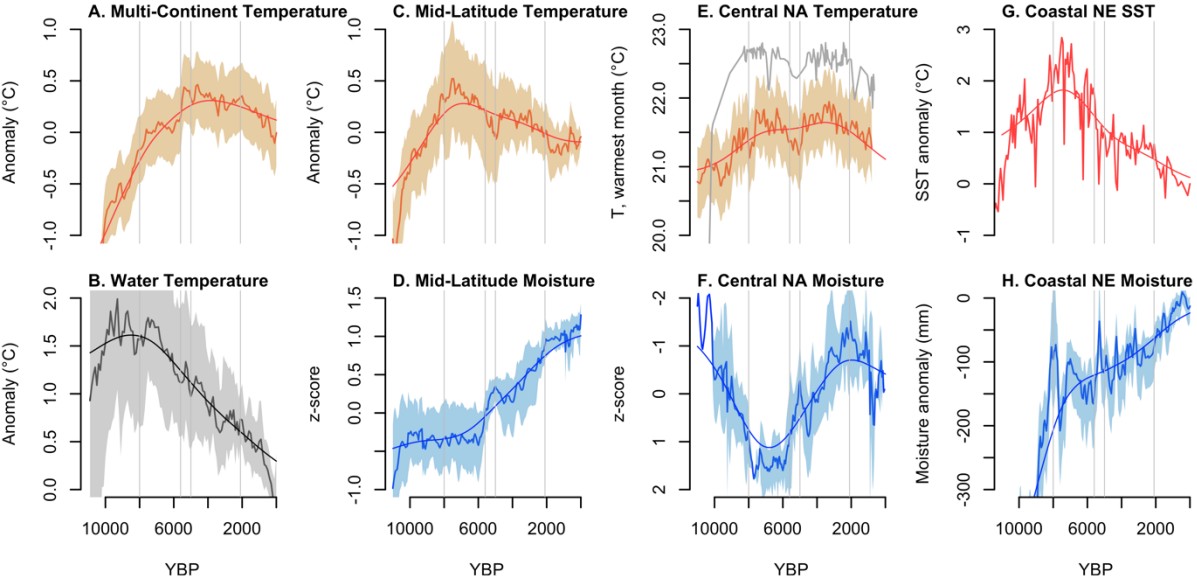

**Figure 3 Holocene paleoclimate time series spanning from multi-continent to local scales representing A) Europe and North America mean annual temperatures (MAT)(Marsicek et al., 2018), B) water temperatures from the same region (Marcott et al., 2013;**

**Marsicek et al., 2018), C) mid-latitude North American mean temperatures of the warmest month (MTWM)(Shuman and Marsicek, 2016), D) mid-latitude hydroclimate indicated by lake-level and dust records (Shuman and Marsicek, 2016), E) central North American MTWM records (Shuman and Marsicek, 2016), F) central North America hydroclimate (dust) records (Dean, 1997;**

Nelson and Hu, 2008; Shuman and Marsicek, 2016), G) alkenone-inferred SST from the Scotian Margin, core OCE326-GGC30 (Sachs, 2007; detrended as in Shuman and Marsicek, 2016), and H) effective precipitation estimated from the lake-level history of New Long Pond, Massachusetts (Newby et al., 2014). In E, the individual MTWM reconstruction from Sharkey Lake, Minnesota (Fig. 1B) is shown (black line) for comparison with the mean of the central sub-region. Vertical lines denote 8000, 5600, 5000, and 2100 YBP. Smooth lines represent a Gaussian smoother used to detrend the records in Fig. 3.

## 3.2 Long-term Holocene trends

In the actual paleoclimate record from within and around North America, both temperature (red, Fig. 3) and moisture (blue, Fig. 3) express long-term trends, although the trends differ by scale, geographic location, season, and climate variable. The waning presence of the ice sheets and their meltwater effects on Atlantic heat transport drove early Holocene warming of >1.5°C (Dyke, 2004; Shuman and Marsicek, 2016), leading to summer temperature maxima from 8000-5500 YBP. Annual temperatures (MAT) did not peak until after 5500 YBP (Fig. 3A) because of the interacting roles of greenhouse gas and winter insolation forcing. Consistent with seasonal insolation forcing (Berger, 1978) and climate model output that demonstrates multi-millennial differences in the timing of temperature maxima based on specific temperature variable considered (Marsicek et al., 2018), sea-surface and lake-water temperatures (Fig. 3B) and mean temperatures of the warmest month (MTWM, Fig. 3C) peaked earlier in the Holocene than MAT. The timing of the summer maximum also varied by sub-region (Fig. 3E, G), in part because of the role of added cen-mil variability. Average water temperatures (gray, Fig. 3B) show a distinct long-term trend because of anomalous Holocene cooling in both alkenone (marine) and chironomid (lake) records, particularly near the western Atlantic (Marsicek et al., 2018; Sachs, 2007); the cooling trends in the water temperatures produces lower temperatures today than during the Younger Dryas (Fig. 3B) and generates the contrast with other datasets and simulations known as the "Holocene Temperature Conundrum" (Liu et al., 2014).

Moisture availability also includes long trends with most areas experiencing higher effective moisture in the late-Holocene than at other times (Fig. 3D, F, H). The mid-Holocene was dry in most areas, but the influence of the Laurentide Ice Sheet before ca. 8000 YBP enhanced moisture availability in central North America (Fig. 3F) and suppressed it substantially along the NE coast (Fig. 3H)(Shuman and Marsicek, 2016). As a result of the opposing directions of regional change, the average moisture available across all of mid-latitude sites remained intermediate until after 5500 YBP when it rose sharply toward present (Fig. 3D). Cen-mil variation modified these trends in different ways in different areas.

## 3.3 Patterns of Cen-Mil Variation

Cen-mil variations rarely departed significantly from the long trends with respect to reconstruction uncertainties in the eight time-series examined here (Fig. 3); detrended series remain near-zero throughout the Holocene (Fig. 4). Detrended temperatures vary only 0.25-0.5°C, which is consistent with Common Era fluctuations (PAGES 2k Consortium, 2013). The rarity of significant anomalies affirms the weak signal-to-noise ratio at cen-mil scales.

Several notable exceptions are apparent, however. A mid-Holocene wet phase from 5600-4500 YBP is the most prominent anomaly. It departs significantly from the mean trends in the mid-latitude, central, and northeastern coastal moisture reconstructions (Fig. 4B-D), although the duration only extends from 5500-5000 YBP along the northeast coast (Newby et al., 2014). Additional multi-century departures appear at 900-700 YBP in the central moisture ensemble (Fig. 4C) and at 4400-4200 and 3400-3100 YBP in the northeast coast (Fig. 4D). (Note that the features recorded in the northeast coast have been replicated in multiple reconstructions; they are not specific to the site or method, Shuman et al., 2019, and are confirmed by detailed radiocarbon dating across multiple cores and sites, Newby et al., 2014, Shuman and Burrell, 2017).

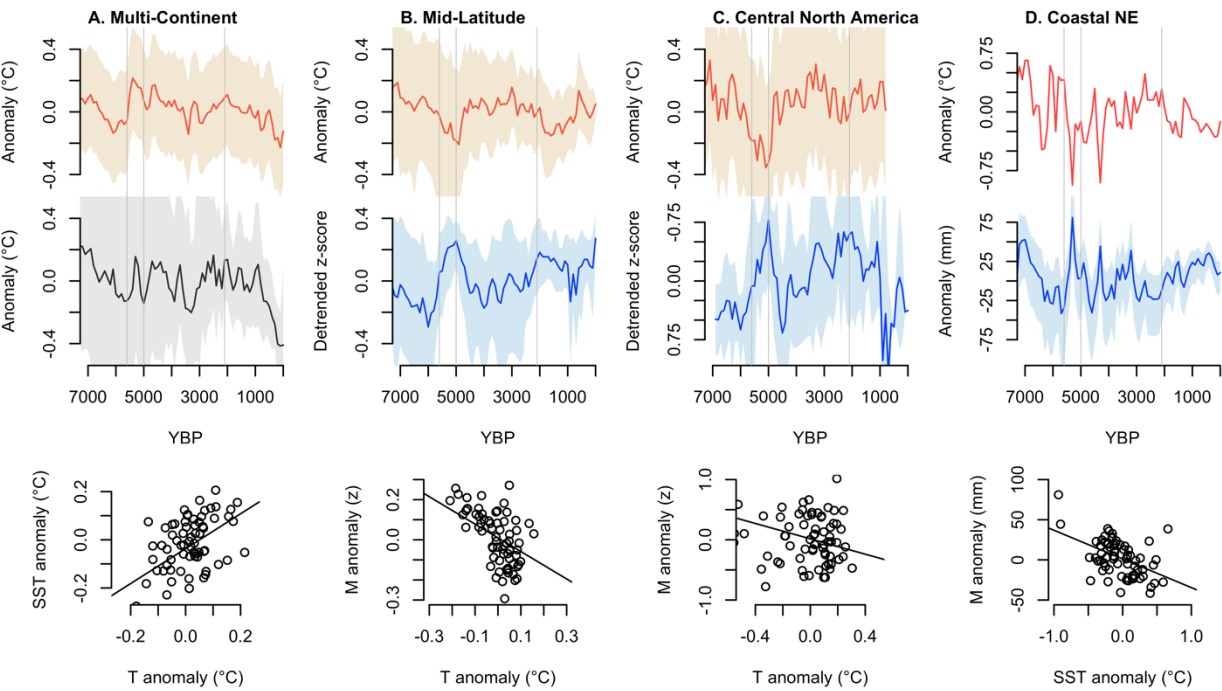

Figure 4. The same paleoclimate time series as in Fig. 3 are shown after detrending. Vertical lines in the upper panels mark 8000, 5600, 5000, and 2100 YBP. Scatter plots show the correlations of each pair of cen-mil residuals for each scale or region over the past 7000 years. Lines in the scatter plots represent generalized least-squares models, which account for temporal autocorrelation. "M" indicates moisture on the y-axes and "T", temperature, on the x-axes of the scatter plots.

Given the small magnitude of most cen-mil variations, correlations among the independent datasets provide support for the rigor of the signals (scatter plots, Fig. 4; Table 1), but not all of these correlations are significant compared to null distributions (Fig. 5). The strongest correlation among two independent time series of cen-mil residuals comes from the mid-latitude scale (Fig. 4B). One time series representing this scale derives from fossil pollen (MTWM, Fig. 4B) and the other from sedimentary evidence of lake-level and dust deposition changes (moisture z-scores, Fig. 4B). Despite the different methods and lakes involved, both mid-latitude datasets represent overlapping geographical distributions (Fig. 1B) and have large, if not

significant, departures from the mean trends at 5600-4500 and 2100-750 YBP (as well as 10,700-9200 YBP apparent in Fig. 3B). Brief departures of the opposite sign appear centered at 750 YBP in both series (Fig. 4B). Pearson's product-moment correlation between the two ensembles equals -0.60 (95% range: -0.42 to -0.73; Table 1), and the correlations exceed those of random surrogates at timescales of 300-1600 years (Fig. 5B). Cooling coincides with moistening at this scale (Fig. 4).

In the coastal northeast sub-region, the cen-mil variation in the SST and lake-level records also correlate significantly (Fig. 4D). Both include repeated multi-century cool and wet fluctuations, particularly at 5500-5000, 4400-4200, 3400-3100, 2100-1300, and 1200-0 YBP. These events alternate with warm and dry departures from the long trend at 4900-4600, 4200-3900, 2900-2100, and 1300-1200 YBP (Newby et al., 2014; Shuman and Marsicek, 2016). The Pearson's product-moment correlation coefficient (r = -0.51, 95% range: -0.66 to -0.31; Table 1) and the slope of the generalized least-squares model (-36±8 mm/°C) both differ significantly from zero (Fig. 4D). The correlations are significant compared to those among random surrogate timeseries at time scales of 100-600 years (Fig. 5D).

At the scale of the central sub-region, the cen-mil correlation between temperature and moisture is weakest (Fig. 4C, 5C). Both ensembles record a large departure from the long-term trends at 5600-5000 YBP, but the other features of the residual series are not correlated (scatter plot, Fig. 4C). The Pearson's product-moment correlation coefficient (r = -0.16, 95% range: -0.38 to 0.08) does not differ significantly from zero, although the slope of the generalized least-squares model does (-0.64±0.22 z/°C) (Fig. 4C; Table 1). The multi-continent scale comparison of MAT and water temperatures (Fig. 4A) also includes correlations that at first appear significant (Table 1), but they are not greater than expected from the range of surrogate random timeseries (Fig. 5A).

**Table 1: Pearson Product Moment Correlations and Generalized Least-Squares (GLS) model slopes, their confidence intervals (95%), and time scales when the correlations exceed 95% of random surrogate timeseries.**

| | Original Citation | r | 97.5% C.I. | 2.5% C.I. | GLS slope (95% CI) | Significant Timescales (yrs) |
|---|---|---|---|---|---|---|
| **Multi-Continent Scale** | | | | | | |
| Mean European and North American Annual Temperatures (°C) | Marsicek et al. (2018) | | | | | |
| Mean Water Temperature (°C) | Marcott et al. (2013) | | | | | |
| | | | | | | |
| | Detrended | **0.57** | 0.71 | 0.38 | 0.72±0.17 °C/°C | NA |

| | | | | | | |
|---|---|---|---|---|---|---|
| **Mid-Latitude Eastern North America** | | | | | | |
| Mean Temperature of the Warmest Month (°C) | Shuman and Marsicek (2016) | | | | | |
| Mean Moisture Index (z-score) | Shuman and Marsicek (2016) | | | | | |
| | | | | | | |
| | Detrended | **-0.60** | -0.73 | -0.42 | -0.68±0.18 z/°C | 300-1600 |
| | | | | | | |
| **Central North America** | | | | | | |
| Mean Temperature of the Warmest Month (°C) | Shuman and Marsicek (2016) | | | | | |
| Mean Moisture Index (z-score) | Shuman and Marsicek (2016) | | | | | |
| | | | | | | |
| | Detrended | **-0.16** | -0.38 | 0.08 | -0.64±0.22 z/°C | NA |
| | | | | | | |
| **Northeast Coastal North America** | | | | | | |
| Uk'37 sea-surface temperature, OCE326-GGC30 (°C) | Sachs (2007) | | | | | |
| Effective precipitation departure from present estimated from lake-level change, New Long Pond, Massachusetts (mm) | Newby et al. (2014) | | | | | |
| | | | | | | |
| | Detrended | **-0.51** | -0.66 | -0.31 | 35±0.7 mm/°C | 100-600 yr |
| | | | | | | |

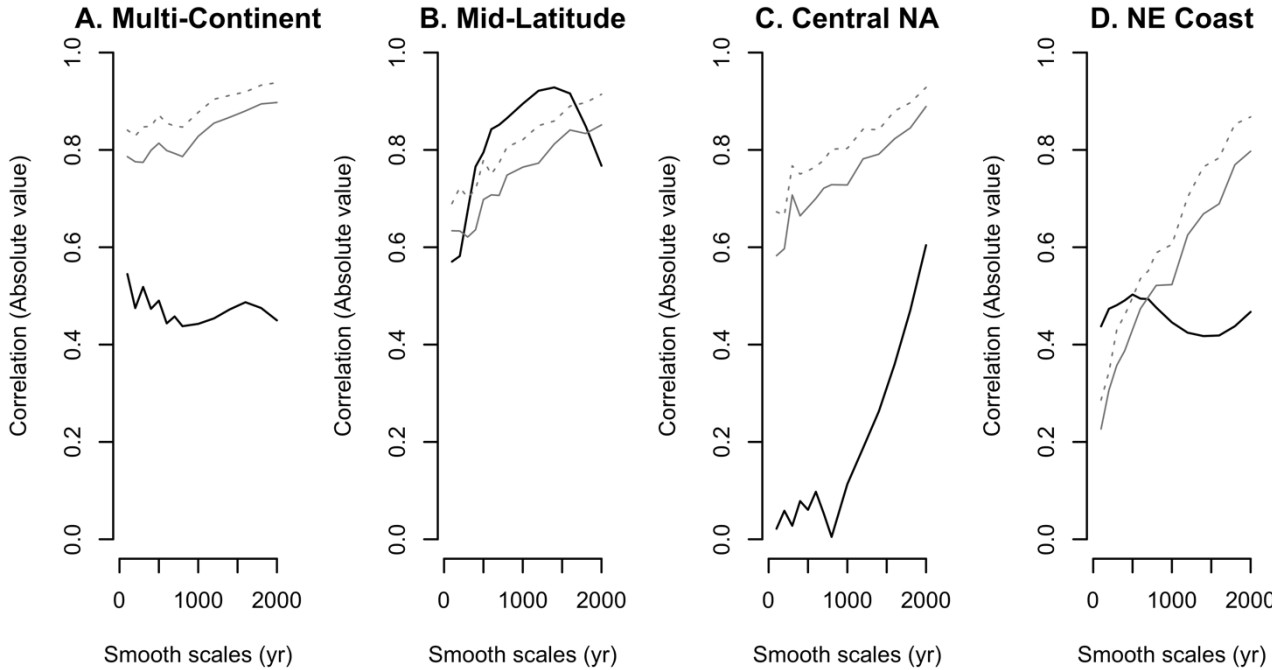

**Figure 5. Timescale dependent correlations for the four spatial scales over the past 6000 years. Black lines show the correlations (as absolute values) between the two detrended paleoclimate time series for each scale or region (as shown in Fig. 3) after Gaussian filtering using cut-off frequencies of 1/t where t (smoothing scale) ranges from 100-2000 yrs; gray lines show the 90% (solid line) and 95% (dashed line) correlations for 1000 surrogate random time series generated as power-law time series with the same spectral slope as the paleoclimate reconstructions using the *GenNullHypPair* function in the corit package (Reschke et al., 2018).**

## 3.4 Spectral analyses

Multitaper spectral analyses of the detrended time series (Fig. 4) show that maximum spectral power exists at the scale of multiple millennia for most of the scales, regions, and variables (Fig. 6). The multi-continent spectra based on land and water temperatures both indicate increasing power from 500 to 5000 yr periods as do the spectra representing the mid-latitude region and central sub-region, which both include substantially more power at near 2000 yr (4000-1500 yr) periods than in the multi-century band. The slopes of the power-law relationships fits to most of the spectra, which are based on z-scores for comparability across variables, range from 1-2, which is 2-4 times steeper than the slopes from the NE coast (Fig. 6).

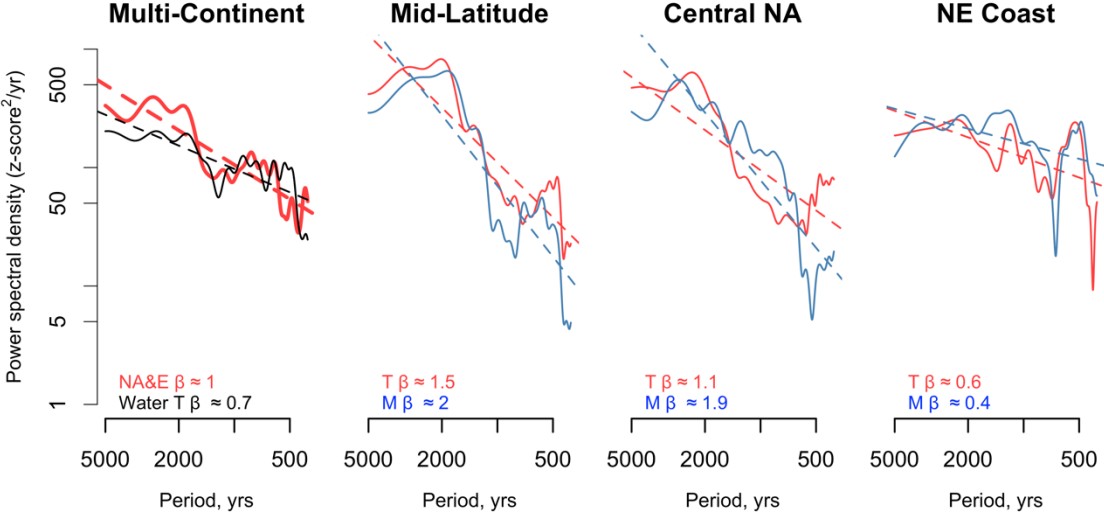

**Figure 6. Multitaper spectral density plots show each time series of detrended cen-mil residuals in Fig. 3, represented as z-scores to enable direct comparison within each pair of independent time series, with respect to log spectral power and period. Beta values represent the slope of the power-law relationship fit to the data. Red curves represent temperature (T) time series and blue curves, moisture (M). Slopes shown here were fit over periods from 5000-400 years.**

The spectra from the NE coastal sub-region differs from the others, but the temperature and moisture spectra from the sub-region are like each other (Fig. 6). Both contain peaks in power at 500-yr periods, which is a timescale of variation evident in the physical stratigraphy of lakes in the region (Newby et al., 2014) and is consistent with the 100-600 year timescale of the significant correlations between the SST and lake-level records (Fig. 5D). The resulting spectral slope of the z-scores is low (~0.6-0.4) because the NE coastal records show greater power in the multi-century band and less power at multi-millennial scales than the mid-latitude region overall and central sub-regions. Thus, the coastal region expresses a higher frequency of variability in the cen-mil bands than the continental areas and larger spatial scales considered here.

### 3.5 Mid-Holocene changes with north-south differences

The differences across spatial scales and regions (Fig. 4) raise the question of the spatial patterns involved. The full ensemble of MTWM reconstructions from mid-latitude sites (Shuman and Marsicek, 2016) enables further evaluation of the most prominent cen-mil feature of the different ensembles, which is the significant mid-Holocene anomaly from ca. 5600-5000 YBP (Fig. 4). To aid the analysis of this anomaly, the MTWM ensemble can be sub-divided according to records that warmed or cooled after the event, which was determined here by subtracting the mean post-event temperature from 4700-4100 YBP from the mean temperature when the anomalies first developed (5600-5000 YBP, Fig. 7). If the temperature difference across these two periods was arbitrary, the timeseries of temperature changes from the warming and cooling regions might expect to show a slow change (i.e., some sites might change early compared to the 600-yr windows, others late; some may change slow, others fast). Instead, rapid changes between 5200-4700 YBP appear in individual records (see e.g., Shuman et al., 2023), the two

subset means (thick lines with uncertainty, Fig. 7A), and temperature-sensitive geochemical records from the two regions (thin lines, Fig. 7A)(Puleo et al., 2020; Henne and Hu, 2010). The two subsets correlate with each other (r = 0.48), but other than the rapid changes at 5200-4700 YBP, the cen-mil features of the two timeseries are not correlated.

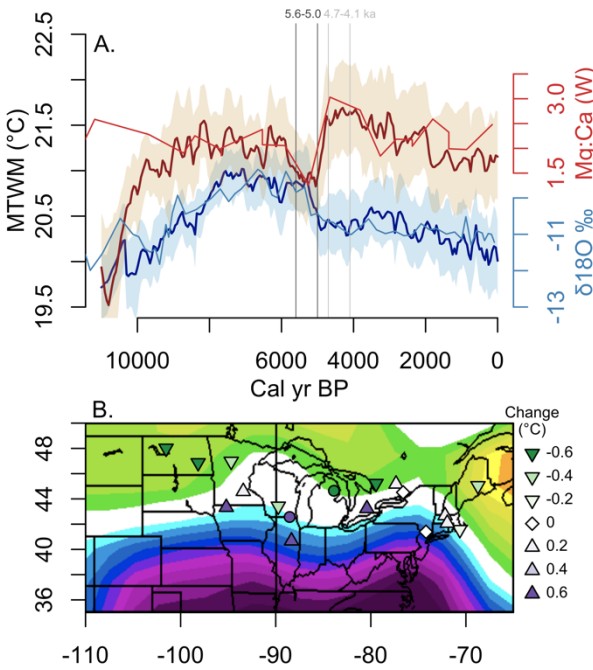

**Figure 7. A) The average time series of mean temperature of the warmest month (MTWM) from sites that either warmed in the mid-Holocene after a cool millennium (red) or cooled after a warm period (blue) are shown along with example temperature-sensitive geochemical records (thin lines) from each region: a reconstruction of water (W) Mg:Ca from Lake Geneva, Wisconsin (purple circle in B)(Puleo et al., 2020) and carbonate oxygen isotope ratios from O'Brien Lake, Michigan (green circle in B)(Henne and Hu, 2010). The 50% range of 100 possible 5-site averages of the MTWM reconstructions is shown as the shaded uncertainty for each series. B) Mapped triangles show the locations of warming (upward triangles) and cooling (downward triangles) based on the differences in the mean temperatures from 5600-5000 and 4700-4100 YBP (denoted by gray and black lines in A, labeled using ages in thousands of years, "ka"). Green shading reflects the magnitude of cooling; purple, the magnitude of warming. Underlying contours indicate historic (1948-2015 CE) correlations between June-August mean temperatures and the North Atlantic Oscillation index from https://www.psl.noaa.gov/data/correlation/; shades of green show positive and purple negative correlations up to 0.4.**

The pattern of change reflects a steepening in the north-south temperature gradient at ca. 5200-4700 YBP (Fig. 7), which additional sites in the northeast U.S. confirm, plausibly also including early cooling in the north before warming in the south (Shuman et al., 2023). However, the mapped changes (Fig. 7B) indicate that the steepening of the temperature gradient was not limited to the northeast U.S. Across the whole mid-latitude region, pairs of sites from different latitudes at any given longitude tend to show rapid warming at the southern site, and cooling at the northern site (Fig. 7B). However, the boundary between areas of warming and cooling does not follow a fixed latitude, but curves across the region as develops today in response to the North Atlantic Oscillation (NAO)(Shuman et al., 2023; Folland et al., 2009)(shading, Fig. 7B). The change also extends beyond the region of a coincident and widespread decline in hemlock (*Tsuga*) pollen percentages, which includes

Ontario and the northeast U.S. (Foster et al., 2006; Bennett and Fuller, 2002), but not the Minnesota, Wisconsin, or other central sub-region records used here. In the northeast U.S., a deviation from the north-south pattern produced a strong coastal-inland contrast of coastal cooling versus inland warming (Marsicek et al., 2013)(Fig. 7B), which is consistent with expected SST changes and resulting coastal cooling on land (Shuman et al., 2023).

## 4 Discussion

### 4.1 Patterns of cen-mil variability

Holocene paleoclimate reconstructions spanning a range of scales from multiple continents to individual sites contain evidence of cen-mil variation within the past 7000 years, after dynamics related to ice sheets and meltwater were no longer significant (Fig. 4). The temperature variations were small (<0.25°C), which challenges detection especially compared to the confidence intervals of the reconstructions (Fig. 3-4). However, reconstruction uncertainty incorporates error associated with reconstructing absolute temperatures, which may not apply to the relative, detrended changes examined here. Furthermore, some of the temperature variations correlate well with independent moisture datasets (Fig. 4-5). The consistency is greater than expected from random variations (Fig. 5), appears evident in the frequency as well as time domain (Fig. 6), and highlights two prominent patterns of cen-mil variation.

First, at the regional scale, mid-latitude MTWM (top, Fig. 4B) and mid-latitude hydroclimate reconstructions (bottom, Fig. 4B) share a correlated pattern of multi-millennial variation (Fig. 5) with spectral power centered at periods around 2000 years (Fig. 6). The broad spectral peak confirms that the variations are not cyclical but produced mid-latitude temperature minima at 5600-5000 and after 2100 YBP (Fig. 4B), which are also evident in both the central (Fig. 4C) and coastal subregions (Fig. 4D). The pattern propagates down to the scale of individual central sites such as Sharkey Lake, Minnesota (black line, top, Fig. 3E). The cool phases coincide with high moisture levels at 5600-5000 and after 2100 YBP, which represent significant departures (non-zero anomalies) from the long-term moisture trends in both the mid-latitude mean (Fig. 3D, 4B) and the NE coastal reconstruction (Fig. 3H, 4D). At least some of the temperature variation appears as a north-south anti-phased pattern across the mid-latitude region (Fig. 7), which bears some similarities to the summer NAO at interannual time scales (Fig. 7B). The most pronounced steepening of the temperature difference between areas that warmed or cooled at ca. 5200-4700 YBP (after the prominent anomaly from 5600-5000 YBP) developed from ca. 4800-3800 YBP (Shuman et al., 2023), possibly indicating different millennial-length phases of negative versus positive NAO states. Whether the phases represent changes in the frequency of interannual NAO modes or millennial-scale variations analogous to, but much slower than, the interannual oscillation is unclear. The pattern has similarities to those detected during the Younger Dryas (Fastovich et al., 2020), but the correlations among temperatures and moisture at cen-mil scales (Fig. 4) differ from the temperature-moisture relationships in the long Holocene trends (Fig. 3), indicating that different processes were likely involved.

Second, ~350-yr variability appears superimposed on the broad-scale variations along the northeast coast (Fig. 4-6). The northeastern SST reconstruction correlates with the drought history along the coast (Fig. 4D), particularly over 100-600 year scales (Fig. 5D), but the high frequency (multi-century) variations appear weaker in most other regions (Fig. 6). A network of northeastern lake-level records also shows evidence of both the millennial pattern evident at the mid-latitude scale and the additional multi-century variability (Newby et al., 2014; Shuman and Burrell, 2017), although the spatial patterns differ from each other and may confirm two superimposed dynamics. The widely prominent event from 5600-5000 YBP includes a strong east-west contrast, but repeated warm-dry events at 4900-4600, 4200-3900, 2900-2100, and 1300-1200 YBP do not (Newby et al., 2014; Shuman and Burrell, 2017; Shuman and Marsicek, 2016). Here, the patterns are detected using independent sets of lakes and cores (i.e., the pollen-inferred temperatures do not derive from the same lakes as the moisture histories).

The first pattern may be stronger at regional scales across mid-latitude North America than across the larger multi-continent scale represented by the MAT reconstruction (Fig. 3A, 4A) because related variations of different sign cancel at the largest scales. The multi-continent MAT and water temperatures include many records in the eastern Atlantic and European regions (Fig. 1A), which may include different influences and expressions; the mid-latitude reconstructions likely emphasize variations common at the scale of westerly waves or other synoptic features, which cancel across regions at the multi-continent scale. The multi-continent MAT record includes a step shift at 5600 YBP (Fig. 3A), which may be related to the prominent mid-latitude anomaly at 5600-5000 YBP (Fig. 4), but sub-regional subsets that represent inland versus coastal areas (Fig. 4C-D) or northern versus southern areas (Fig. 7) highlight the added geographic differences.

In central North America, cen-mil variation appears weaker and less consistent than in other areas (Fig. 4-5). Potentially, teleconnections and interactions in this region may be more variable than in other areas, such as the northeast coast. The difference is analogous to different teleconnected responses to tropical Pacific and North Atlantic variability today, which have more consistent responses to interannual variability in some areas than others.

**4.2 Mid-Holocene Anomaly**

The most significant anomaly in many records falls between 5600-5000 YBP (Fig. 3-4). Qualitative evidence of a "Mid-Holocene Anomaly" has existed for decades in North America. At Elk Lake, Minnesota, the varve record contains a phase of reduced dust deposition and varve thicknesses (Dean, 1997). Fossil pollen records from the Great Plains, such as Creel Bay, Spiritwood, and Moon lakes, North Dakota, and Pickerel Lake, South Dakota, contain a conspicuous interval of low *Ambrosia* (ragweed) pollen (Grimm, 2001), while unique ostracode assemblages at 5600-5000 YBP indicate severe drought in Minnesota (Smith et al., 2002). Lake-level records in the Rocky Mountains indicate a distinct phase of high water in Wyoming and Colorado (Shuman et al., 2014; Shuman and Marsicek, 2016), while those in inland areas of the northeast U.S. record low water at the same time (Newby et al., 2011; Shuman and Burrell, 2017). Submerged tree stumps within Lake Tahoe, California, date to this interval (Benson et al., 2002), which also includes anomalies in other regions (Magny et al., 2006; Magny and

Haas, 2004), such as the Mediterranean Sea (Alboran Cooling Event 2; Cacho et al., 2001; Fletcher et al., 2013) and Africa
(Berke et al., 2012; Thompson et al., 2002).

A similar mid-Holocene anomaly has been widely recognized across the North Atlantic region. Records document shifts in deep water from 5500-4700 YBP (Oppo et al., 2003) and other distinct anomalies from the Labrador to Norway in sea-surface temperatures, salinity, loess and precipitation (Giraudeau et al., 2010; Jackson et al., 2005; Larsen et al., 2012; Orme et al.,
2021). An earlier dust accumulation anomaly on the Greenland summit, interpreted to represent a shift in atmospheric circulation from 6000-5100 YBP, may be related to these patterns (Mayewski et al., 2004). Likewise, a negative phase of the NAO may be consistent with distinct and contrasting phases of anomalous warmth on the Labrador Shelf from 5700-4800 YBP (Lochte et al., 2020) and cold in the Florida Strait from 5500-4400 YBP (Schmidt et al., 2012). The weak latitudinal temperature gradient before the transition at ca. 5000-4700 YBP was followed by a steep gradient until ca. 3500 YBP,
particularly in the northeast U.S. and the adjacent Atlantic (Shuman et al., 2023), and could be consistent with either a shift from negative- to positive-dominated NAO regimes or an analogous low-frequency change in the Atlantic pressure field. However, the sign of the NAO analog may depend upon whether the patterns detected here represent mean annual or summer temperature shifts (contrast Fig. 7B with similar maps for mean annual anomalies in Shuman et al., 2023).

The mid-Holocene anomaly from 5500-5000 YBP, like a possible anomaly of the opposite sign after 4800 YBP (Fig. 7B)(Shuman et al., 2023), would not be expected to appear in all records or regions. Spatial heterogeneity in detecting the pattern would be consistent with the small magnitude of cen-mil variations (Fig. 4), interactions with other trends (Fig. 3), and the inherent spatial variability of such dynamics (Fig. 7). In some areas, the millennial-scale variation could at first delay and then reinforce the direction of the long-term trends producing a step change (e.g., northern sites in blue in Fig. 7A). Such a
step-like cooling pattern may extend across eastern Canada and northern and eastern Greenland, which could be consistent with NAO-like outcomes in annual temperatures (Briner et al., 2016; Shuman et al., 2023). Only in some areas, where the millennial variation temporarily counteracted the long trends, would the anomaly be expressly apparent without detrending (e.g., southern sites in red in Fig. 7A; moisture variations in central North America, Fig. 3F).

### 4.3 Potential drivers

The dynamics driving the two patterns of cen-mil variation here (broad-scale millennial variations including at 5500-5000 YBP and northeast coastal multi-century variations) need further investigation. As noted above, the north-south differences across mid-latitude sites may indicate a role for variability in the Atlantic pressure gradient similar to the NAO at monthly to interannual time scales. NAO-like dynamics may be expressed at cen-mil scales (Olsen et al., 2012; Orme et al., 2021; Shuman et al., 2023) and are usually strongest in winter, but they can have an important form of expression in summer, extending from
North America to Europe (Folland et al., 2009). The prominence of the mid-Holocene shift in the mid-latitude records examined here (Fig. 6-7) and elsewhere (Willard et al., 2005) also raises the question of whether the change originated from

other feedbacks that affect Atlantic sector pressure gradients such as by altering the depth of the African monsoon low (Claussen et al., 1999), potentially via dust loading feedbacks (Pausata et al., 2016), or whether millennial-scale variation interfered with long trends to produce state-shifts on multiple continents at approximately the same time. For example, northern cooling by ca. 4700 YBP (Fig. 7), which may reflect NAO-like dynamics that began as early as 6200 YBP in some areas (Shuman et al., 2023), could have been a driving factor in the chain of dynamics that produced rapid changes to the African monsoon system (Collins et al., 2017). Resulting feedbacks may then have reinforced the abrupt changes at mid- and high latitudes (Muschitiello et al., 2015)(Fig. 7).

The multi-century variability expressed along the Atlantic coast also has similarities to variations observed in other Atlantic sectors, including off the west African coast (Adkins et al., 2006; deMenocal et al., 2000) and the Mediterranean (Fletcher et al., 2013). Holocene simulations indicate Atlantic-African-American linkages should be expected (Muschitiello et al., 2015). These sectors may be linked via the dynamics of the North Atlantic subtropical high (Clement et al., 2015), possibly driven by cloud feedbacks (Bellomo et al., 2016), atmospheric feedbacks on meridional overturning (Wills et al., 2019) or volcanism (Birkel et al., 2018; Kobashi et al., 2017); similar dynamics on interannual scales have consequences for North American hydroclimate patterns (Enfield et al., 2001; Anchukaitis et al., 2019).

A preliminary comparison of alkenone records from the Scotian and Virginia margins (Sachs, 2007) and the west African margin (Adkins et al., 2006) reveal that they potentially share a multi-century pattern of variability over the past 6000 years, which is expressed by different sign changes between Scotian and African margins with the sign of the variation switching between early and late events off Virginia (Fig. 8). Principal components analysis indicates that, at face value given current age control, the pattern represents 11% of the variance within the three records over the past 8000 years, but 23% in the past 6000 years. (The first PC represents 72% of the variance and the long-term trend over the past 8000 years). The multi-century SST variability correlates with the coastal hydroclimate variability (Fig. 8) and may also involve shifts in the temperature gradient over the western Atlantic and Greenland (Shuman et al., 2019). However, the multi-century SST and hydroclimate variability along the western Atlantic margin does not extend to similar large-scale temperature gradient changes (Fig. 7). Therefore, at least two different dynamics, even if they are different seasonal expressions of NAO-like variations, must be involved.

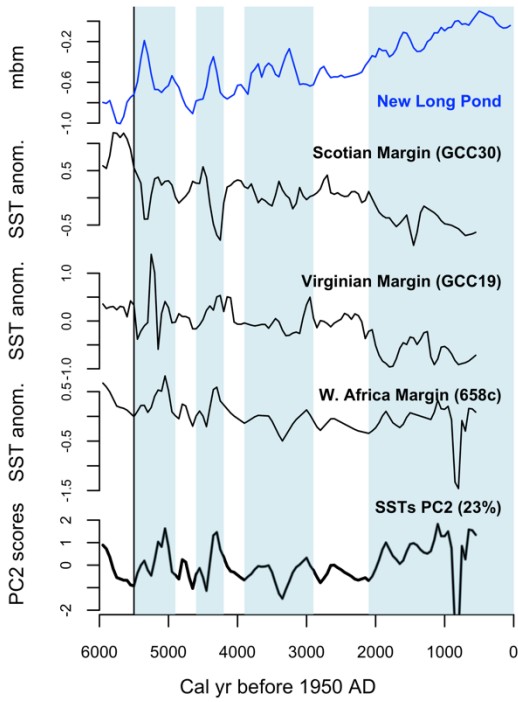

**Figure 8. Comparison of the lake-level record from New Long Pond, Massachusetts (Fig. 3H), and three alkenone-inferred sea-surface temperature (SST) records (Adkins et al., 2006; Sachs, 2007). The SST records are shown after detrending and with the second principal component (PC2) scores of a PCA conducted on the non-detrended series; the first PC captures the trends. Blue bands represent the timing of wet phases identified from radiocarbon dated paleoshorelines at multiple lakes in coastal**
**Massachusetts (Newby et al., 2014). "mbm" refers to meters below modern water level.**

### 4.4 Signal detection

Cen-mil variability is small relative to the ability to detect it. Several factors may aid efforts to examine these dynamics. First, analyses of calibrated rather than relative paleoclimate records may aid detection if all records involved in an analysis capture

the variations in similar, linear ways (e.g., tree-rings, Anchukaitis et al., 2019; pollen, Shuman et al., 2023; marine geochemical records, Osman et al., 2021). For example, calibration of fossil pollen using the modern analog or similar techniques removes inherent non-linearities in the responses of individual plant taxa and isolates the effects of focal climate variables from the interacting influences of other factors. While the responses of individual taxa are non-linear, the calibrated pollen assemblage (plant community) response is likely to be linear (Shuman et al., 2019). Other calibrated indices, such as chironomid- or

alkenone-derived temperatures, should behave similarly, although relative paleoclimate indices may be more complex (Axford et al., 2011). Relative indices may be particularly problematic if they are not multi-variate and only shift in a bivariate fashion with respect to a wide range of different influences (e.g., sediment carbon:nitrogen ratios, stable isotopes). For example, some isotope data series may not be meaningfully different from random noise in some settings if multiple competing influences,

such as temperature, airmass trajectories, precipitation seasonality, evaporation, and groundwater inputs, interact to create complex signals (e.g., Donovan et al., 2002; Steinman et al., 2010). However, systematic treatment of stable isotopic datasets based on their relationship to the locally dominant driving processes can avoid this type of problem (Konecky et al., 2023).

Second, ensembles of detailed, well-dated records from the same climatic region or scale increase the likelihood that noise can be removed through averaging (Fig. 2)(Konecky et al., 2023; Shuman et al., 2023), although doing so may also reduce the amplitude of the cen-mil variations (sd ratios <1 in Fig. 2B)(Hébert et al., 2022). Such ensembles may need to be created and analyzed in a manner that accounts for a) the potential that different climate variables or regions can change asynchronously at the beginning and ends of events (Fig. 7A)(Rach et al., 2014; Gonzales and Grimm, 2009; Ma et al., 2012) and b) different expressions of the same events in different regions or sub-regions (Fig. 7B)(Shuman et al., 2023; Fastovich et al., 2020). These issues can be resolved by using cluster analyses or similar techniques to sub-divide potential sets of records (Shuman et al., 2023) and focusing on timeseries correlation, especially compared to surrogates (Reschke et al., 2019), or the temporal overlap of full events rather than precise synchrony of the abrupt changes that bound them (Parnell et al., 2008; Newby et al., 2014).

Third, comparisons of independent ensembles of data from the same area allow the signals to be cross validated. Analyses here benefitted from ensembles based on many cores (>600 fossil pollen records, >30 cores across nine lake-level study sites, dozens of water temperature records), which reduced age and reconstruction uncertainties. The different ensembles of temperature and moisture were also developed in fundamentally different ways (e.g., microfossil versus physical sedimentological analyses), which indicated that any shared signals had a broad basis of evidence indicating multivariate environmental responses to even weak climate fluctuations. Finally, and even in the absence of the above criteria, comparisons of paleoclimate reconstructions with random surrogate datasets (Fig. 5) provides a powerful tool for distinguishing between spurious and significant variations (Fig. 2)(Reschke et al., 2019; Telford and Birks, 2011).

## 5 Conclusions

In areas centered on mid-latitude eastern North American, coherent cen-mil variation has similar signal magnitudes in independent temperature and hydroclimate datasets. The major signals, including prominent mid-Holocene anomalies from 5600-5000 YBP, are not readily falsified using the analyses here because they share the characteristics of non-random signals. The variations played important roles in mediating long-term trends, creating differences in the timing of the Holocene thermal maximum and rates of early-Holocene warming and late-Holocene cooling among regions. Events that have previously attracted focused attention, such as at 9300, 8200, 4200, and 2700 YBP, do not stand out as the most distinct features of the ensembles of data compiled here.

Instead, two major patterns of variability appear evident. One appears expressed across a broad range of spatial scales, which is similar to some unforced global variability in transient climate simulations (Marsicek et al., 2018; Wan et al., 2019). The pattern includes a set of millennial fluctuations with important anomalies and rapid shifts in both temperature and moisture gradients at ca. 5000 and 2100 YBP. More work is needed to evaluate whether such broad-scale changes, including mid-Holocene anomalies, stem from external forcing, such as solar or volcanic events, intrinsic ocean or atmosphere variability, or

other factors such as surface-atmosphere feedbacks, but the spatial patterns indicate that shifts in summer atmospheric pressure gradients over the Atlantic, such as associated with the NAO at short time scales, may be involved. The second major pattern produced a series of multi-century hydroclimate fluctuations along the Atlantic coast and may relate to Atlantic SST variability with distinctive spectral properties not seen in most of the study area. Because pollen and lake-level records detect the events, they had ecological, hydrological, geomorphic, and therefore, likely human significance and deserve further investigation.

## 6 Data availability

The different ensembles of data examined here are available through the NOAA National Centers for Environmental Information Paleoclimatology Database:

- North Atlantic and adjacent continental reconstructions (Marsicek et al., 2018): https://www.ncdc.noaa.gov/paleo-
575 search/study/22992
- Mid-latitude North America and sub-regional temperature and moisture reconstructions (Shuman and Marsicek, 2016): https://www.ncdc.noaa.gov/paleo-search/study/31097
- New Long Pond, Massachusetts, USA (coastal northeast U.S.) lake-level reconstruction (Newby et al., 2014): https://www.ncdc.noaa.gov/paleo-search/study/23074
- Scotian Margin sea-surface temperature reconstruction (OCE326-GGC30) (Sachs, 2007): https://www.ncdc.noaa.gov/paleo-search/study/6409

## 7 Acknowledgements

This work was supported by funding from U.S. National Science Foundation (DEB-1856047; EAR-1903729). C. Routson, C.
585 Morrill, L. Curtin, D. Groff, I. Stefanescu, R. Hébert, and an anonymous reviewer kindly provided thoughtful comments on the manuscript.

The author declares that he has no conflict of interest.

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

.