# Peer review of "Patterns of Centennial-to-Millennial Holocene Climate Variation in the North American Mid-Latitudes"

_Climate of the Past, 2022_

## Referee Comment (RC1)

This manuscript presents a framework for evaluating climatic signals at millennial to centennial timescales in Holocene paleoclimate data, to overcome the challenge of discerning between signal and noise in reconstructions. The author demonstrates how the structure of synthetic datasets changes in a scenario with or without a prescribed signal; these changes can then help us differentiate between signal and noise in actual paleoclimate data. Across a variety of spatial and temporal scales, two excursions and/or patterns of climate signal are identified: 1) millennial temperature and moisture fluctuations between 5500 and 2100 on many spatial scales; and 2) multi-centennial hydroclimate fluctuations along the Atlantic coast. The author proposes that further work should be devoted to investigating the mechanisms driving these fluctuations.

Overall, I think the manuscript presents a useful potential solution to an important problem in our community. Similar methods could be widely applied in paleoclimate data compilations, to either confirm or disprove the existence of fluctuations in paleo-data that are attributed to climatic changes, even if they might occur spuriously. My comments mainly concern the presentation of results; with so many datasets, spatial scales, and temporal scales of interest, I sometimes found myself a bit lost when trying to draw out the major conclusions from all these analyses on all these regions / scales. Prior to publication, I hope Dr. Shuman will consider the following suggestions, which I believe would strengthen the analysis and conclusions:

1. **Spatial scale visualization:** I found that I had to re-read the descriptions of the different spatial scales of interest (L72-80), which then becomes a bit more confusing when the author refers to "continent" and "mid-continent" and "mid-latitude" (the latter two of which correspond to "regional" and "sub-regional" I believe?) etc. I think a series of maps as an introductory figure would be very helpful. As it stands, we don't get to visualize the spatial scales we're looking at until Fig. 6, which feels a bit of an afterthought. As an alternative to having the reader pull up the referenced studies repeatedly, it might be nice to have a multi-panel figure that shows a) the spatial extent of each different scale / dataset, and b) the records included in each.

2. **Concise presentation of findings across spatial / temporal scales:** Again, I found it difficult to follow the presentation of results with so many different, but similar sounding, regions and timescales considered. I think it would be helpful to have a table that summarizes the results; a similar concept to what's presented in Fig. 4, but a bit more easily digestible. For example, the reader could quick look to see whether there are significant relationships between X variable and Y region and Z timescale. I think it's reasonable to have something similar to Fig. 4 but with fewer acronyms, more words, and overall less burden on the reader to quickly pull key findings. You could also integrate information about other aspects of the study such as power spectra, potential drivers / mechanisms of recognized shifts, etc.

3. **Broad relevance**: The author explains early in the manuscript that he is using the chosen datasets as an illustrative example of these methods, and then in Sec. 4.4 (Signal detection) mentions that it might be challenging to apply this framework to other types of proxy records (e.g., isotope records that are more often (compared to pollen) influenced by multiple climate variables). Rather than simply stating this, I think the paper would be more broadly useful if the author proposed some ways to overcome multi-variate issues that might be encountered in other proxy types. As it stands, this would appear to be less an illustrative example of how the method can be applied, and more an argument that this

can only be done with very specific types of paleoclimate data. So, if possible, I suggest closing the paper with some forward looking suggestions gleaned from this effort.

A few minor editorial comments:
L35: start list with a colon and separate components using semicolons
L152: "signal to noise" instead of "sign to noise"
Fig 2: it's somewhat confusing that European data (here and elsewhere) is included in a paper about North America; maybe explicitly state why this is done (presumably either because its important for capturing Atlantic-related variability, or because its already a feature of the existing datasets), or remove?
L522: "millennial" instead of "millennia"

---

## Author Response (AR1)

Dear Professor Reyes,

I appreciate the patience with the revision process and value the insights from the reviewers. Both reviewers seemed to understand the main goals of the manuscript and supported the main conclusions, and I have tried to follow the spirit of their suggestions to enhance the paper. Their reviews tighten the findings of paper by suggesting new figures, changes in the analyses, and textual clarification.

Regarding review #1, their critiques of the manuscript make sense to me. In considering a revision, I have tried to be more systematic in the naming of the different regions and datasets, and rephrased the wording to make each distinct. I have also developed a site map (new Fig. 1) to show the different regions involved and have updated the text to explain why European data are sometimes included and discussed, such as in line 76 of the Introduction and in line 130 of the Methods. I have replaced Figure 4 (scatter plots) with a table of the main relationships and statistics (Table 1). Finally, Section 4.4 has been revised to better address the broad applicability of the approaches discussed here, including by citing some other recent papers and the methods suggested by review 2. I have also tried to make the minor corrections suggested, although I have also removed some text in response to the reviews and that negated the suggestions.

Regarding review #2, I am very appreciative of the comments from Dr. Hébert. They have been stimulating and helpful. In particular, I am glad to be pointed toward up-to-date methods that can be applied to determine the significant patterns. I had been planning to cite some of these newer papers (e.g., Hébert et al., 2022) even before seeing the review, but I have re-run my analyses incorporating a Gaussian filter in place of the LOESS, developing random surrogate timeseries using 'corit', and used them to test the significance of the correlations as described by Reschke et al. (2019). I had been trying to do a simple version of this approach and was glad to revise with such a well-designed and validated method. Likewise, I re-interpolated the datasets to 100-yr timesteps to reduce concerns about oversampling the underlying data at 50-yr intervals. The changes in the analyses created modest changes to the result, but overall, help to solidify the primary conclusions.

I have also made the following additional revisions:
1) provided a more formal definition of cen-mil variations at the start of section 2 (Methods);
2) removed Figure 4, and replaced it with a new table of statistical results, given the agreement among reviewers about making this figure more concise;
3) updated and clarified Figure 5 (now Figure 6 because of new added figures) by removing the shading and slope uncertainties;
4) adding a new Figure 5 that shows the timescales of significant correlations among the different datasets, which should help address concerns about the meaning of spectral power at the ~500-yr band in the revised spectral analysis figure (Fig. 6);
5) clarified (in lines 343-349) that the raw data clearly show ~500-yr variability at the northeastern coastal sites;

6) removed most references to the standard deviation ratios because I agree that the averaging across records should reduce the signal amplitudes as shown by Hébert et al., 2021;

7) clarified that the spectra in the updated Fig. 6 (previous Fig. 5) were calculated using z-scores to make the temperature and moisture reconstructions directly comparable;

8) simplified Figures 6-7 into a single new Figure 7, which includes a color scale and focuses on the main mid-Holocene shift rather than correlation across the entire pair of timeseries;

9) addressed concerns that arbitrarily selecting any time period and dividing the data into positive and negative change groups (in Fig. 7) would create spurious anomaly patterns by a) showing the patterns of change in selected individual records and b) re-calculating the direction of change using the average difference between two 600-yr periods rather than a fixed change point; by doing so, I tried to minimize the risk of a spuriously constructed abrupt change, which would be smoothed over the 600-yr windows and consequently reduced in amplitude (if it were spurious);

10) further clarified that the time period examined in Fig. 7 was not selected arbitrarily but was found previously to represent the largest rate of change in multiple proxies in eastern North American data over the past 8000 years (line 140 in the Methods) and citing (in lines 375-380) a more in-depth analysis of the spatial patterns of this change in a separate dataset with a greater number of records: Shuman, B. N., Stefanescu, I. C., Grigg, L. D., Foster, D. R., & Oswald, W. W. (2023). A millennial-scale oscillation in latitudinal temperature gradients along the western North Atlantic during the mid-Holocene. *Geophysical Research Letters*, 50, e2022GL102556. https://doi.org/10.1029/2022GL102556;

11) removed references to minor or insignificant changes (e.g., 8200-yr event in line 358) in the datasets and focus on the major cen-mil patterns.

Overall, I have updated the analyses to be more consistent with the state-of-the-art analyses cited by Dr. Hébert, while retaining some of the explanatory discussion of random variations to help illustrate the need to use methods such as the null distributions of correlation coefficients to test for significance of (typically weak) cen-mil variations in paleoclimate datasets. I have also tried to correct all of the smaller issues that were noticed.

Thank you for the opportunity to submit a revision.

Sincerely,
Bryan Shuman

---

## Referee Report (RR1)

Dear Alberto Reyes,

I have reviewed the revised manuscript with interest and acknowledge the great work that the author put in to take into account my comments and suggestions. The author presents varied evidence and a stimulating discussion of centennial to millennial scale variability that I think should be published in Climate of the Past once the minor points I raised below have been considered.

Additionally, I wish to apologize to the authors for the delay in my review as I was off academic duties the last few months (and am generally slow to review in addition).

Cordially,
Raphaël Hébert

Minor Points:
Line 170: Weird sentence bit "such as dominates"
Line 188: I don't think smoothing is the correct term, because smoothing inter-annual variability can only lead to less variability and not create low-frequency one. Something like "integrate" or "accumulate" would be more appropriate. I would also avoid characterizing it as "white noise" since inter-annual variability is not generally white noise, albeit it can in fact be quite close to it especially in more continental regions.
Line 190: In this case I agree smoothing processes can generate artificial low-frequency variability by interaction with proxy processes, just not real one.
Line 212: I'm confused how the regional comparison is spanning eastern and central NA, whereas the sub-regional, which should be smaller, compares central and coastal NA, which appears similar to me as we are comparing central NA with either eastern NA or coastal NA.
Line 228: Typo "an potential"
Line 275: I'm not convinced that this is sufficient to avoid all biases, but I guess it should decrease potential biases.
Line 320: Provide reference for the "previously published confidence interval".
Line 368: Unclear to me what is done here. We have data that is already binned/interpolated at 50- or 100-year resolution, so are the 50-year ones re-binned at 100-year by averaging two nearby points? Or are we using a Gaussian smoother to reinterpolate? I guess I don't understand what is meant by we smooth and detrend at 100-year resolution.
Line 375: Were the surrogates also detrended with the Gaussian filter? I'm just thinking that this might bring down the correlations in the surrogate on the longest timescales on Figure 5 and potentially make some of the results more significant, e.g. for Figure 5B, the drop in correlation in the midlatitude series could be because the low-frequencies have been detrended whereas the surrogates were not and keep increasing. I'm not sure it would make a difference though as the filter is 6000 years and so maybe it doesn't actually impact 2000-year timescales, but might be worth checking.
Section 1.2 I don't see the difference between null distributions and null expectations, so maybe this section could be integrated with the previous discussion of null distribution, although I see that the two are serving different purposes, this section is rather an example that isn't used for hypothesis testing but for a pedagogical demonstration, so it could also be titled accordingly and kept separate.

Line 385: Do null expectations aid detection? Or rather they allow to assess whether the detection is significant or not.

Line 596: Maybe it would be good to give the reconstruction uncertainties. It makes one wonder though what is the significance of this variability if the reconstruction uncertainties are higher. However, the errors on the reconstructions are generally not independent from each other in a given series. In the case of pollen reconstructions for example, we may take the RMSE of the calibration database as a measure of the errors related to the transfer function, but this will include a component that will be related to predicting the absolute temperature value and likely the same for all the samples, such that it will be taken out when detrending and looking at anomalies with respect to the trend. I don't know how we can separate the two types of errors though, I just wanted to note that the part of the reconstruction errors that is independent between the sample might not be so big compared to the anomalies, but again, I'm not sure how we can begin to separate them formally.

Line 629: I'm just wondering here whether the two datasets are truly independent, or whether there are records extracted from the same lakes or cores.

Table 1: Formatting could be improved, there is a lot of blank space in the table. I could see it reduced to 4 rows.

Figure 5: Typo in "Raeske", should be "Reschke"

Figure 6: I assume the slopes were fit over the entire range? One thing to think about is that the detrending is going to remove low-frequency variability, so it could be an idea to compare the detrended and undetrended spectra and see where the two diverge in order to see when the power loss occur and maybe not fit it. It depends what we want to measure with this, it the long-term trend is say a forced component that is removed with independent information, then we might say that the residuals represent the internal variability and that the result is the slope of the internal variability. In this case however, the trend removes everything, so the result is a fit of the real variability with a bias low from the long timescales dominated by the power loss.

Section 3.5 Is there no point in giving the correlation between the two groups of series?

Line 958: This is an interesting idea that NAO would manifest on longer timescales. Just a question of wording here, but I think it would be better to say "millennial-length phases dominated by either negative or positive NAO states" to make clear that it just means that one phase may occur more frequently than the other, but they would still both occur on inter-annual timescales right.

Line 1117: And here it could be written "negative- to positive-dominated NAO regimes" or something like that. I'm not sure what's the best way, but I think it would be useful to make a distinction given the different timescales involved.

---

## Author Response (AR2)

Dear Dr. Reyes,

I greatly appreciated the final review comments and have tried my best to make the appropriate changes throughout the manuscript.

Sincerely,

Bryan Shuman

Response to specific review comments:

Minor Points:
Line 170: Weird sentence bit "such as dominates"
- Changed to "such as is common on"

Line 188: I don't think smoothing is the correct term, because smoothing inter-annual variability can only lead to less variability and not create low-frequency one. Something like "integrate" or "accumulate" would be more appropriate. I would also avoid characterizing it as "white noise" since inter-annual variability is not generally white noise, albeit it can in fact be quite close to it especially in more continental regions.
- Replaced "smooth" with "integrate" and changed "white noise" to "stochastic"

Line 190: In this case I agree smoothing processes can generate artificial low-frequency variability by interaction with proxy processes, just not real one.

- I modified the wording to leave in "smoothing", but also refer to "integrating"

Line 212: I'm confused how the regional comparison is spanning eastern and central NA, whereas the sub-regional, which should be smaller, compares central and coastal NA, which appears similar to me as we are comparing central NA with either eastern NA or coastal NA.

- I see how the wording was confusing here. I revised to refer to the 'region' as 'mid-latitude North America' and the sub-regions as "the central and northeastern (NE) subregions of mid-latitude North America"

Line 228: Typo "an potential"
- Corrected 'an' to 'a'

Line 275: I'm not convinced that this is sufficient to avoid all biases, but I guess it should decrease potential biases.
- This makes sense. I tried to clarify some wording here, but I also don't have a great way to fully avoid the biases. However, I have updated Fig. 7 to show that the patterns in the sub-sets developed using this approach usefully anticipate patterns in geochemical datasets from the two regions.

Line 320: Provide reference for the "previously published confidence interval".
- The citations were added and the text clarified to explain that this refers to the 8 datasets examined throughout the paper.

Line 368: Unclear to me what is done here. We have data that is already

binned/interpolated at 50- or 100-year resolution, so are the 50-year ones re-binned at 100-year by averaging two nearby points? Or are we using a Gaussian smoother to reinterpolate? I guess I don't understand what is meant by we smooth and detrend at 100-year resolution.

- Revised to clarify that the 100-yr intervals have been selected from all datasets for the smoothing and detrending analysis.

Line 375: Were the surrogates also detrended with the Gaussian filter? I'm just thinking that this might bring down the correlations in the surrogate on the longest timescales on Figure 5 and potentially make some of the results more significant, e.g. for Figure 5B, the drop in correlation in the midlatitude series could be because the low-frequencies have been detrended whereas the surrogates were not and keep increasing. I'm not sure it would make a difference though as the filter is 6000 years and so maybe it doesn't actually impact 2000-year timescales, but might be worth checking.
- The surrogates were generated from the beta slopes of the detrended data, which I have clarified in the text.

Section 1.2 I don't see the difference between null distributions and null expectations, so maybe this section could be integrated with the previous discussion of null distribution, although I see that the two are serving different purposes, this section is rather an example that isn't used for hypothesis testing but for a pedagogical demonstration, so it could also be titled accordingly and kept separate.

- I retitled the section "Demonstration of random effects"

Line 385: Do null expectations aid detection? Or rather they allow to assess whether the detection is significant or not.

- Revised the wording to state that "Null expectations can help assess whether analyses show significant patterns."

Line 596: Maybe it would be good to give the reconstruction uncertainties. It makes one wonder though what is the significance of this variability if the reconstruction uncertainties are higher. However, the errors on the reconstructions are generally not independent from each other in a given series. In the case of pollen reconstructions for example, we may take the RMSE of the calibration database as a measure of the errors related to the transfer function, but this will include a component that will be related to predicting the absolute temperature value and likely the same for all the samples, such that it will be taken out when detrending and looking at anomalies with respect to the trend. I don't know how we can separate the two types of errors though, I just wanted to note that the part of the reconstruction errors that is independent between the sample might not be so big compared to the anomalies, but again, I'm not sure how we can begin to separate them formally.

- I have struggled with this same issue and have not been clear on how to best address it. However, I wanted to be clear here that the RMSEs usually used as the reconstruction uncertainty are large compared to the apparent signals. I have revised and included a sentence trying to summarize the issue.

Line 629: I'm just wondering here whether the two datasets are truly independent, or whether there are records extracted from the same lakes or cores.

- I have added a sentence to clarify that they are truly independent datasets, derived from different sets of lakes. (In an earlier related study, Shuman et al., 2019, the comparison was made in the same lake, but using different cores from different water depths with different age models).

Table 1: Formatting could be improved, there is a lot of blank space in the table. I could see it reduced to 4 rows.

- I agree. I will address this issue as part of typesetting if the manuscript is accepted.

Figure 5: Typo in "Raeske", should be "Reschke"

- I corrected the caption.

Figure 6: I assume the slopes were fit over the entire range? One thing to think about is that the detrending is going to remove low-frequency variability, so it could be an idea to compare the detrended and undetrended spectra and see where the two diverge in order to see when the power loss occur and maybe not fit it. It depends what we want to measure with this, it the long-term trend is say a forced component that is removed with independent information, then we might say that the residuals represent the internal variability and that the result is the slope of the internal variability. In this case however, the trend removes everything, so the result is a fit of the real variability with a bias low from the long timescales dominated by the power loss.

- I added a sentence to the caption clarifying that "Slopes were fit over the full range of periods from 5000-400 years." I appreciate the additional suggestion for exploring the patterns here but am not able to fully evaluate this issue within the scope and timeframe of the revision.

Section 3.5 Is there no point in giving the correlation between the two groups of series?

- I revised section 3.5 to incorporate the geochemical records now shown in Fig. 7 and have added a sentence about the correlations.

Line 958: This is an interesting idea that NAO would manifest on longer timescales. Just a question of wording here, but I think it would be better to say "millennial-length phases dominated by either negative or positive NAO states" to make clear that it just means that one phase may occur more frequently than the other, but they would still both occur on inter-annual timescales right.

- I am uncertain whether the changes represent changes in the frequency of NAO modes or millennial-scale anomalies analogous to, but much slower than, the interannual oscillation. I added a sentence to make this point.

Line 1117: And here it could be written "negative- to positive-dominated NAO regimes" or something like that. I'm not sure what's the best way, but I think it would be useful to make a distinction given the different timescales involved.

- Good suggestion. I modified the wording to reflect the potential time scales involved.